# Addressing mechanism bias in model-based impact forecasts of new tuberculosis vaccines

M. Tovar [1,2], Y. Moreno [1,2,3] & J. Sanz [1,2] ✉

In tuberculosis (TB) vaccine development, multiple factors hinder the design and interpretation of the clinical trials used to estimate vaccine efficacy. The complex transmission chain of TB includes multiple routes to disease, making it hard to link the vaccine efficacy observed in a trial to specific protective mechanisms. Here, we present a Bayesian framework to evaluate the compatibility of different vaccine descriptions with clinical trial outcomes, unlocking impact forecasting from vaccines whose specific mechanisms of action are unknown. Applying our method to the analysis of the M72/AS01$_E$ vaccine trial -conducted on IGRA+ individuals- as a case study, we found that most plausible models for this vaccine needed to include protection against, at least, two over the three possible routes to active TB classically considered in the literature: namely, primary TB, latent TB reactivation and TB upon re-infection. Gathering new data regarding the impact of TB vaccines in various epidemiological settings would be instrumental to improve our model estimates of the underlying mechanisms.

Despite the decay in TB incidence and mortality achieved worldwide since 1990[1], its yearly rate of reduction is arguably too slow to meet the goal settled by the World Health Organization (WHO) in the End-TB strategy, which consists of completing a reduction of TB incidence and mortality rates by 90% and 95%, respectively, between 2015 and 2035[2]. Instead, starting in 2020, we are witnessing, for the first time in decades, an alarming increase in global TB burden levels with respect to previous years, with as many as 1.6 million casualties attributable to TB worldwide in 2021, combining HIV negative and positive cases (1.5 and 1.4 in 2020 and 2019, respectively[3]). The cause of this increase was the irruption of the COVID-19 pandemic, which threatens, in countries like India, to raise the TB death toll back to even higher levels in the next few years[4]. This issue, as well as the ever-increasing rates of emergence of drug resistance[5], evidence the need of new epidemiological interventions and tools against TB; paradigmatically a new and better vaccine than the current bacillus Calmette-Guerin (BCG)[6], whose efficacy against the more transmissible respiratory forms of the disease in young adults is disputed[7].

Vaccine testing for TB is especially difficult due to a number of factors. They include the slowness of the contagion dynamics that forces vaccine developers to consider studies involving larger numbers of participants during longer follow-up periods than for other diseases[8–10], as well as the difficulty in defining trial endpoints for a disease where infection status can only be ascertained indirectly, and immunological correlates of protection remain elusive[11]. This makes the testing of TB vaccines an extremely challenging and expensive task, in spite of which, nowadays, several preventive vaccine candidates against TB are being tested in human clinical trials[6,12]. Some candidates have completed phases 1, 2, and 2b of their development, and are about to enter into phase 3 to test their efficacy at providing prevention of infection (PoI) and/or prevention of TB disease (PoD) in large cohorts of thousands of participants recruited in high-burden settings. In this context, the first phase 2-2b trials to be completed for new preventive vaccines against TB were those of the candidates MVA85A[8], M72/AS01$_E$[9,10], and also H4:IC31[13], which was compared to a revaccination protocol with BCG (BCG-revac). These candidate vac-

[1]Institute for Biocomputation and Physics of Complex Systems (BIFI), University of Zaragoza, Zaragoza 50009, Spain. [2]Department of Theoretical Physics, University of Zaragoza, Zaragoza 50009, Spain. [3]Centai Institute S.p.A, 10138 Torino, Italy. ✉e-mail: jsanz@bifi.es

cines, which collected disparate efficacy readouts, were tested within trials of noticeable diverse designs in several key characteristics such as geographical distribution, participants' age, or IGRA status, as detailed in Table 1.

The comparison summarized in Table 1, involving just three pioneer phase 2/2b efficacy trials for novel tuberculosis vaccines, suggests, given the diversity of their designs, that the question of what is an optimal strategy for testing a preventive vaccine against TB at these stages of vaccine development lacks a unique answer, and that, as the rest of the vaccine candidates progresses through the development pipeline, the field will witness a higher number of trial designs being explored, as anticipated in[14]. This multiplicity of trial designs, along with the paucity of resources to allocate for evaluating novel TB vaccine candidates at a global scale[15], makes it absolutely critical to ensure that vaccines with different target product profiles, and, or estimated from trials of different characteristics can be timely compared in their expected ability to halt the global epidemics of TB.

One of the reasons why such a task is difficult is the fact that the PoI or PoD efficacy readouts obtained from an efficacy trial do not offer an unequivocal characterization of a TB vaccine, since the same risk-reduction readouts observed in a trial can be mapped onto different mechanisms of action in different vaccine candidates[14]. This is extremely important because some of these compatible mechanisms are impossible to distinguish just by interpreting the trial's results using standard methodologies, and yet, they appear associated with significantly different impacts, as foreseen by transmission models, if applied in simulated vaccination campaigns[16].

In this work, we propose a Bayesian modeling approach in order to relax such kind of assumptions. In our framework, we define a family of possible compartmental vaccine models characterized by different vaccine mechanisms from each of which we can estimate the likelihood associated with a particular trial outcome. Using those likelihoods combined with uniform, non-informative priors for each of the possible models in the family, we can estimate the posterior probabilities of each model, providing in this way a means to evaluate the compatibility of each of the possible models with the outcome observed in a specific trial. Finally, we use these Bayesian posteriors as natural weights for each model's impact forecasts, which -at least within the breadth of the family of models considered- do not depend on mechanistic assumptions anymore.

To illustrate our approach, we analyze the case of the multicentric clinical trial of the candidate vaccine M72/AS01$_E$, conducted on IGRA-positive individuals from settings in three different high-burden countries: Kenya, Zambia, and South Africa, which led to a promising PoD readout of $VE_{dis} = 49.7\%$ (95% CI $2.1 - 74.2$). Specifically, we apply our formalism to evaluate the a posteriori plausibility of the different vaccine descriptions that can be built as all-or-Nothing vaccine models[17–19] by incorporating in their parametrizations different combinations of protective effects. Furthermore, we identify the specific combinations of protection mechanisms that generate model descriptions that are more plausible, under the light of the observed trial outcome. This offers a rationale for selecting the most adequate vaccine model structures, and weight them in order to produce mechanism-agnostic impact forecast averages.

## Results

In a trial such as the one carried out for the vaccine M72/AS01$_E$, conducted among TB−, IGRA+ individuals without a past of active TB, the episodes of incident TB to be observed during the study can be divided into three different groups, or routes to disease. First, some of the individuals whose IGRA conversion had occurred relatively recently will be expected to progress to primary TB during the first 12–24 months after exposure to the pathogen, which will typically overlap with the follow-up period. This happens at a fast progression rate denoted as $r$ in this study (see Fig. 1A). Second, enrolled

**Table 1 | Phases 2/2b clinical trials for new TB vaccines**

| Vaccine candidate | Trial ID (year) (1) | IGRA status | Number of participants (2) | Follow-up duration (median) | Participants age | Location | Efficacy readout(s) (+95% CI) |
|---|---|---|---|---|---|---|---|
| MVA85A | NCT00953927 (2009-2011) | Negative | 1395 (Placebo) 1399 (MVA85a) | 24.6 | 4–6 months | South Africa | PoI: −3.8% (−28.1–15.9) PoD: 17.3% (−31.9–48.2) (5) |
| M72/AS01E | NCT01755598 (2014-2015) | Positive | 1660 (Placebo) 1623 (M72/AS01E) (3) | 32.4 | 18–50 years | South Africa, Zambia, Kenya | PoD: 49.7% (2.1–74.2) (6) |
| H4:IC31 BCG-revac | NCT02075203 (2014-2015) | Negative | 306 (Placebo) 297 (H4:IC31) 312 (BCG) (4) | 24 | 12–17 years | South Africa | PoI (H4:IC31): 30.5 % (−15.8–58.3) PoI (BCG-revac): 45.4% (6.4–68.1) (7) |

(1) Years of the recruitment phase. (2) Individuals included in efficacy analyses. (3) Individuals in according-to-protocol efficacy cohorts. (4). Individuals in per-protocol analyses. (5) PoD efficacy corresponds to endpoint definition 1, as described in ref. 8. (6) As reported in the final analysis[10], for the according-to-protocol efficacy cohort, first definition. (7) Efficacy against sustained QFT conversion (QFT conversion without reversion within 6 months).

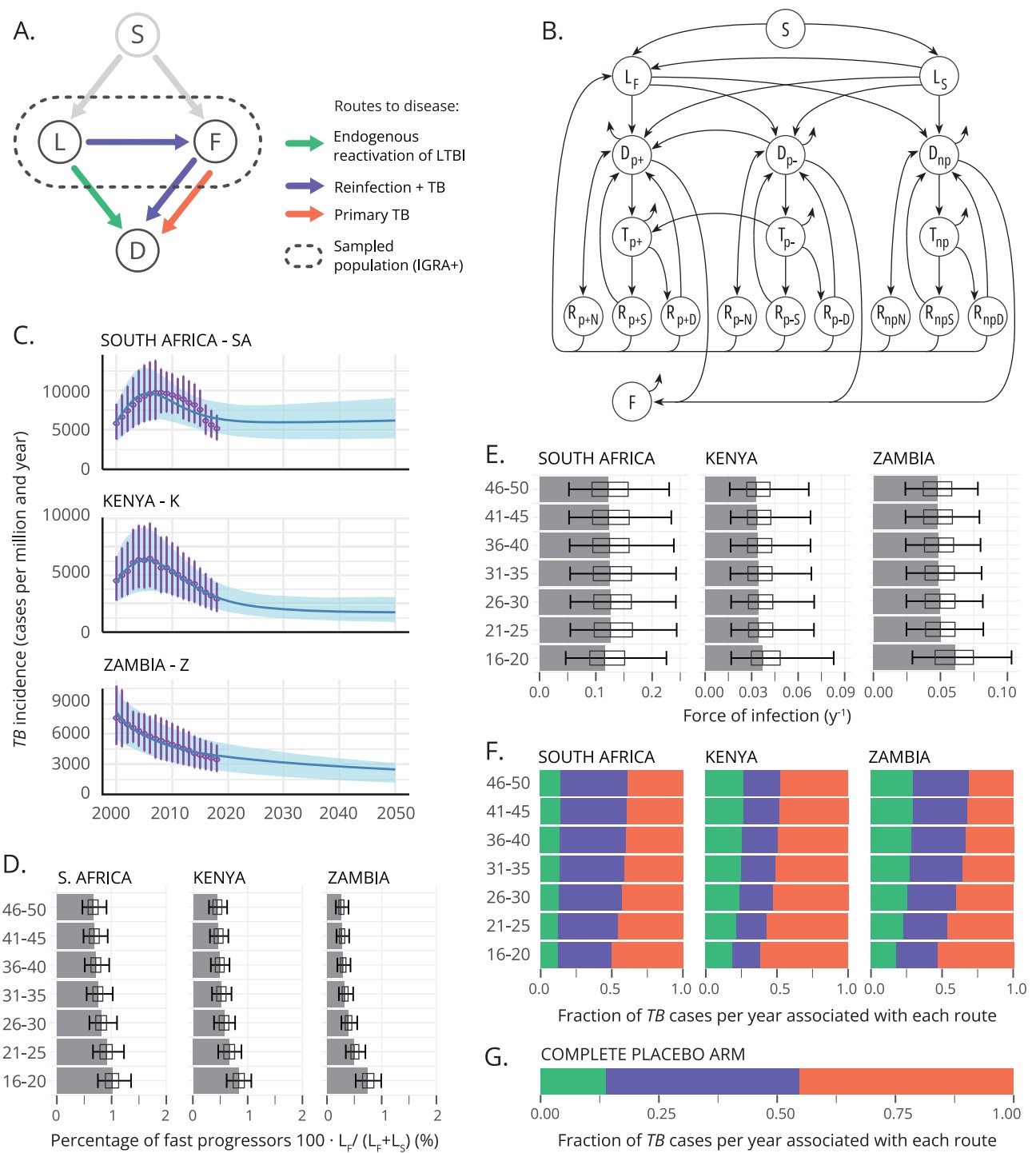

individuals whose IGRA+ status is associated with a latent TB infection (LTBI, linked to an exposure occurred, typically, >2 years before the beginning of the study) would be at a much lower risk of experimenting endogenous reactivation during the trial, mapped to a slow transition rate denoted as $r_L$ in Fig. 1A, with $r_L \ll r$. Third, enrolled individuals may undergo primary TB followed upon re-exposure to the pathogen during the study, which, in Fig. 1A happens at a rate proportional to $q \cdot \beta \cdot p$, where $\beta$ means the basal force of Infection, and $q$ is a reduction coefficient capturing the relative risk of infection of previously infected (IGRA+) with respect to unsensitized individuals (IGRA −). These three possible routes to active TB, sketched in the compartmental model diagram in Fig. 1A, are classically referred to as the "three risks model"[20], a frame coined by Vynnycky and Fine in 1997[21]. In

Fig. 1A we distinguish each of them according to one of the most commonly assumed model structures found in TB modeling literature[22], where LTBI individuals are split into fast (F) vs slow progressors (L), which we have chosen to describe the transmission dynamics of the placebo arms considered in this study.

Leaning on this basic model description of disease dynamics sketched in Fig. 1A, our first goal is to implement computational simulations to estimate the relative weight of each route to disease in the incidence observed in the placebo arm of a clinical trial such as the M72/AS01$_E$ study. To implement such simulations, we need two main ingredients: the epidemiological parameters $r$, $r_L$, $q$, and $p$ governing the transitions, as well as the expected initial prevalence of fast (F) vs. slow progressors (L), and the forces of infection $\beta$ in the population

**Fig. 1 | A priori characterization of the three routes to disease in the placebo arm of a Phase 2b clinical trial conducted on IGRA+ participants.**
**A** Compartmental model used to describe TB dynamics in the placebo arm of a trial conducted on IGRA+ individuals without past or present evidence of active TB. According to this model, trial participants can be divided in fast (F) vs. slow (L) progressors, each of which show different risks of progression to disease (D) per unit time that can be further divided into three routes to disease. **B** Compartmental model used to describe TB transmission at country-level. Individual states are: susceptible $S$, infected (either fast $L_F$ or slow $L_S$ progressors, analogous to F and L reservoirs in **A**); active disease $D$, disease under treatment $T$ (pulmonary smear-positive TB (p+), pulmonary smear-negative TB (p−) and non-pulmonary TB), disease recovery $R$ (Natural ($_N$), successful after treatment ($_S$), treatment default ($_D$)) and treatment failure ($_F$). The model is used first to obtain estimates of parameters to inform clinical trial simulations, and later to evaluate vaccine impact (for further details see Supplementary text S1 and ref. 23). **C** The country-level model sketched in **B** is calibrated to reproduce TB incidence trends (also mortality, see Supplementary Fig. S1) in each country. Error bars represent the reported uncertainty of

incidence estimates in the WHO tuberculosis database, shaded areas capture the 95% CI in all the trajectories forecasted by the model ($N = 500$). **D** From the calibrated simulations conducted country-wise, we retrieve estimates for the relative fraction of fast progressors over the total population of IGRA+ individuals without a past of active TB $L_F/(L_F + L_S)$ that is **e**xpected in each country. **E** From the same simulations, we obtain estimates of the force of infection per country and age group (fraction of susceptible individuals infected per year and age group). **D**, **E** bars represent the median, boxes capture the inter-quartile range, and error bars represent the 95% CI from a set of $N = 500$ simulations. **F** With those items at hand, along with literature-based estimates for the epidemiological parameters $r$, $r_L$, $q$, and $p$, the placebo arm of the M72/AS01$_E$ study can be simulated in-silico, from which we can estimate the expected fraction of incident TB cases associated to each of the routes to disease. **G** Weighting the contributions estimated in **F**, according to the age and country-wise distributions of participants in the M72/AS01$_E$ trial[9], we obtain an overall estimate of the relative contribution of each route to disease to the total incidence observed in the global placebo arm of the trial. Source data are provided as a Source Data file.

sampled during the recruitment phase of the clinical trial. While the epidemiological parameters $r$, $r_L$, $q$ and $p$ are extracted from previous literature (see Methods), we resort to computational modeling to gather age-specific estimates of the relative weight of F and L reservoirs, as well as of the force of infection. More specifically, in order to obtain estimates of these parameters we use the same TB transmission model that will be later used to evaluate the impact of the vaccine, whose compartmental structure is sketched in Fig. 1B. This model, first introduced in ref. 23, can be conceived as a more comprehensive version of the model represented in Fig. 1A, designed not to provide a description of the disease transitions within the context of a trial, but to provide an exhaustive description of TB transmission dynamics on the whole population of an entire country during several decades. The model is based on a system of ordinary differential equations (see Methods and supplementary text S1 for further details), and integrates information from assorted bibliographic sources (epidemiologic parameters), demographic databases (the UN population division database, from where time-evolving demographic structures are extracted), as well as from the WHO TB database (TB incidence and mortality trends) around a detailed age-structured description of TB dynamics. Calibrating the force of infection and the diagnosis rates, this model reproduces TB incidence and mortality trends reported in the WHO Database in the three countries of the M72/AS01$_E$ study (South Africa, Kenya, and Zambia, Fig. 1C) during the period 2000–2018. From this calibration procedure, we then obtain a complete model-based description of the dynamical evolution of TB epidemics in each country. Specifically, from its results we extract the desired estimates of the relative prevalence of fast (F) vs slow (L) individuals in the population (Fig. 1D), as well as the estimates of the basal force of infection $\beta$ in each country (South Africa, Kenya and Zambia) and age group, both of them evaluated in 2015 (Fig. 1E), the year that the M72/AS01$_E$ study took place (see Methods and Supplementary text S1 for further details).

With those ingredients at hand, we then perform a first set of in silico trial simulations stratified per age group in each of these three countries, wherein participants' fates are simulated stochastically, according to an implementation of the Gillespie algorithm that allows tracking the three routes to disease independently (see Methods and Supplementary text S1). Through these simulations, we quantify the fraction of total TB cases associated with each route to disease in the placebo arms enrolled in each country, stratified per age group (Fig. 1F). In the multi-centric trial of M72/AS01$_E$, participants between 18 and 50 years old (724 between 18 and 25 years old; 321 between 25–30, and 594 between 30.50) were enrolled in South Africa (80.8%), Kenya (14.9%), and Zambia (4.3%)[9,10]. Considering these demographics, we produce a global estimate of the contribution of each route to disease to the incidence observed in the placebo arm of the entire

study in the M72/AS01$_E$ trial, as an average of the results of the age groups and countries involved in the study, weighted by their relative frequencies (Fig. 1G).

Now, within the framework of the "three risks model", it is conceptually possible that vaccines may provide PoD by reducing only the disease risk associated with some of these three routes to disease. The estimates of the relative share of total incidence that can be attributed to each route to disease give us very useful information about what are the precise mechanisms that may be more interesting to target in a given population. However, the immunological components of host responses against *Mycobacterium tuberculosis* (*M.tb.*, the causative agent of TB) that are involved in protecting against primary TB upon recent infection, endogenous reactivation, or re-infection are complex and neither homogeneous nor linear;[24] and could be boosted to different extents by a vaccine in a way that is difficult to predict a priori.

To accommodate modeling decisions to this uncertainty, we consider a set of vaccines that provide PoD by reducing each of the three individual risks, either alone, or combined (Fig. 2A). This yields seven vaccine models that can be denoted as $M(i,\varepsilon)$, where we have each model being defined by the integer index $i \in \{1,2,\ldots,7\}$, determining the specific protection mechanism(s) present in the vaccine (Fig. 2A) and the continuous parameter $\varepsilon \in [0,1]$, which captures the intrinsic efficacy, modeled as the fraction of individuals protected within an all-or-nothing modeling framework, considered identical for all the vaccine effects present in each case. This way, while in models 1–3 only one of the three routes to TB is disrupted by the vaccine, models 4–7 incorporate several mechanisms simultaneously (Fig. 2). For instance, model 5 describes a vaccine that protects against primary TB and against LTBI endogenous reactivation at the same time, and model 7 represents a vaccine tackling all three routes alike. Crucially, as represented in Fig. 2B, the maximum fraction of total TB cases that a vaccine behaving according to each of these models can prevent is variable, spanning from 13.8% of cases that would be prevented by a vaccine with a 100% efficacy against LTBI reactivation only, to the obvious 100% of cases, that would be prevented by a perfect vaccine with 100% efficacy against all routes of TB alike.

Therefore, the set of vaccine descriptions $\{M(i,\varepsilon)\}$ will constitute a space of possible models, within which we will look after the one(s) whose assumptions are most compatible with a trial's PoD readout of vaccine efficacy, that is, with the largest Bayesian posteriors, instead of blindly assuming that a vaccine acts through a given specific mechanism, or, for example, that it reduces all risks alike. To accomplish that task, we expand the Gillespie stochastic simulations mentioned before to include the simulation of vaccine arms each of them coherent with the seven types of vaccine models described. Specifically, we assume a uniform non-informative prior on the efficacy parameter, and register the observed efficacy against disease $VE_{dis}$

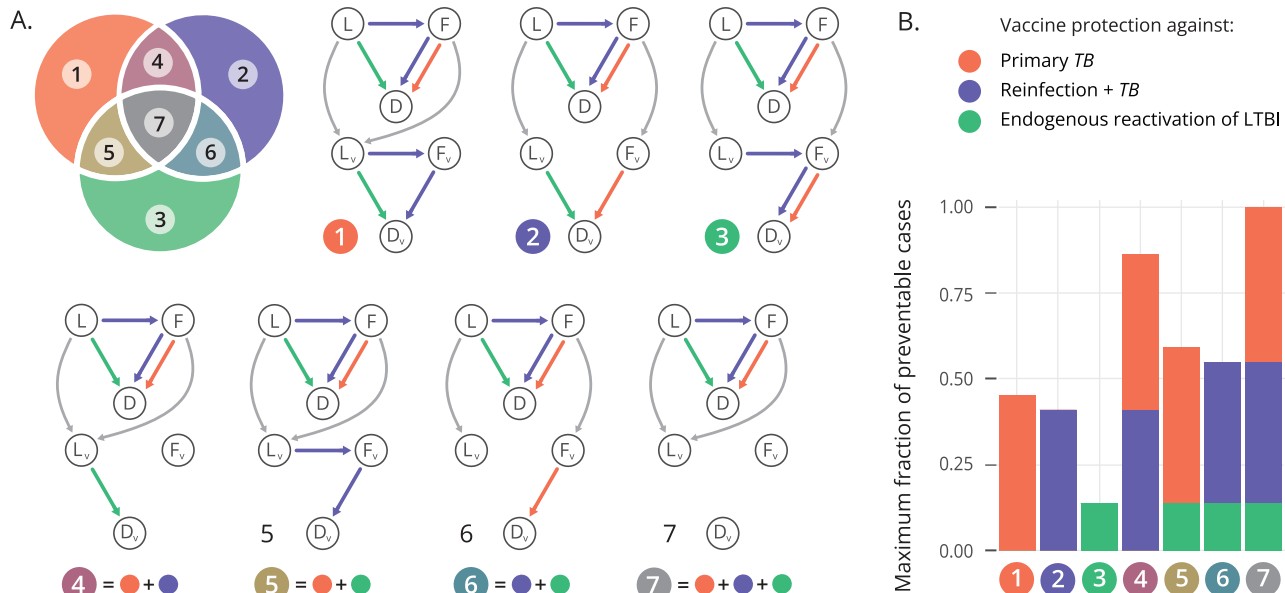

**Fig. 2 | Compartmental models to accommodate the description of vaccines providing PoD by acting on specific routes to disease. A** (Top left): Venn diagram sketching the seven vaccine types contemplated in the study, as a function of the routes to disease they are assumed to protect against: primary TB (model 1), TB upon re-infection (model 2) or endogenous reactivation of LTBI (model 3). Combinations of these mechanisms yield models 4–7, which describe vaccines that are able to halt two (models 4, 5 and 6), or all three routes to disease at once. Each region of the Venn diagram corresponds to a value of the discrete index

$i \in \{1, 2, \ldots, 7\}$. Beside the diagram, we show the compartmental descriptions of each vaccine type. In each of the seven models, a vaccine arm is included in parallel to the placebo arm, that defines the disease dynamics of the vaccinated individuals who are protected against developing disease through the corresponding routes. **B** Maximum fraction of preventable cases by each vaccine, considering the mechanisms of protection present in each case. Source data are provided as a Source Data file.

(that is, the PoD readout) that is associated with each trial simulation (see Fig. 3A). After we conduct a total amount of two million simulations for each model type, we consider the likelihood $P(VE_{dis} = 49.7\%|i,\varepsilon)$ associated with each possible model defined by the combination of parameters $\{i,\varepsilon\}$. Integrating these likelihoods over all possible values of $\varepsilon$ for each vaccine type we retrieve the marginalized likelihood curves $P(VE_{dis} = 49.7\%|i)$ represented along with the simulation clouds in Fig. 3A. Red dashed lines on top of these marginalized likelihoods capture the probability that a trial with the specifications of the one described in ref. 9,10, conducted on a given vaccine behaving according to the i-th vaccine model, will lead to a PoD efficacy readout $VE_{dis}$ that is compatible with the observations made in the real trial: $VE_{dis} = 49.7\%$[10].

Using this marginalized likelihood, we can apply Bayes rule to define the marginal posterior probability associated with each particular model, $P(i|VE_{dis} = 49.7\%)$. These marginal posteriors $P(i|VE_{dis} = 49.7\%)$, represented in Fig. 3B, provide a mean to quantify the relative support in a given trial's outcome for each one of the seven different vaccine descriptions provided. In our case, the observed PoD efficacy readout reported for the vaccine M72/AS01$_E$[10] appears more compatible with models 4, 5, 6, or 7, each featuring a combination of several vaccine mechanisms, than with models where vaccine effects are associated to a unique mechanism of action. The reason behind this emerging hierarchy between vaccine models is the relation between the observed $VE_{dis}$ and the maximum fraction of events that are preventable by each type of vaccine (shown in Fig. 2B). Posteriors of models whose maximum fraction of preventable cases is smaller than the observed $VE_{dis}$ are in turn smaller, meaning that the protection mechanisms present in these vaccines are likely insufficient to explain the observed trial result (models 1, 2 and 3: see Figs. 2B, 3A). In what regards the remaining vaccine models (4 to 7), all of them provide protection against several routes to disease, featuring maximum fractions of preventable cases that are well above the $VE_{dis}$ value observed in the trial (see Fig. 2B). Their different posteriors can then be

understood by comparing the relative frequency at which each model is expected to generate simulated values for $VE_{dis}$ that are compatible with the trial observation, when all simulations at all possible values of $\varepsilon$, distributed around the uniform, non-informative prior, are considered (marginal density curves over the $VE_{dis}$ axis in Fig. 3). Observing those curves, -the marginalized likelihoods, integrated over $\varepsilon$ for each model as defined in Eq. 4 (see Methods)- we see that models 4–6 show marginalized densities with a tighter spread around intermediate values of $VE_{dis}$ than model 7, showing higher values around the observed $VE_{dis} = 49.7\%$, and therefore higher model posteriors than model 7. Maximum fractions of preventable cases for models 4, 5 and 6 are smaller than that of model 7, which is equal to 1, since the latter can potentially prevent all TB cases by blocking all routes to disease alike. This translates into a cloud of simulated data for model 7 with a steeper slope in Fig. 3A, which in turn causes the flatter marginal density curve for model 7 than for models 4–6. Taken together, these results lead to the slightly lower marginal posterior probability observed for model 7 than for models 4–6, evaluated at $VE_{dis} = 49.7\%$ (Fig. 3B)

Our approach can also be used to estimate the intrinsic efficacy values $\varepsilon$ that are most compatible with the observed PoD efficacy readout $VE_{dis} = 49.7\%$ under each vaccine model, by applying the Bayes rule over each of the seven types of models independently to obtain the conditional posteriors $P(\varepsilon|VE_{dis} = 49.7\%, i)$. The first momentum of these conditional posteriors corresponds to the expected values of the intrinsic efficacy parameter under each model type, that is $\langle\varepsilon\rangle_i$, which is captured in Fig. 3C, along with its confidence intervals obtained from the conditional posterior distribution $P(\varepsilon|VE_{dis} = 49.7\%, i)$ itself. As expected, these efficacy estimates illustrate a sensible feature of our model approach, namely, that a given PoD readout $VE_{dis}$ must be mapped to lower intrinsic efficacy $\varepsilon$ values when the vaccine is able to halt progression to disease through all possible routes than when it acts on a subset of them.

Once we have described our Bayesian approach to inform vaccine characterization combining trials' results with in-silico simulations, we

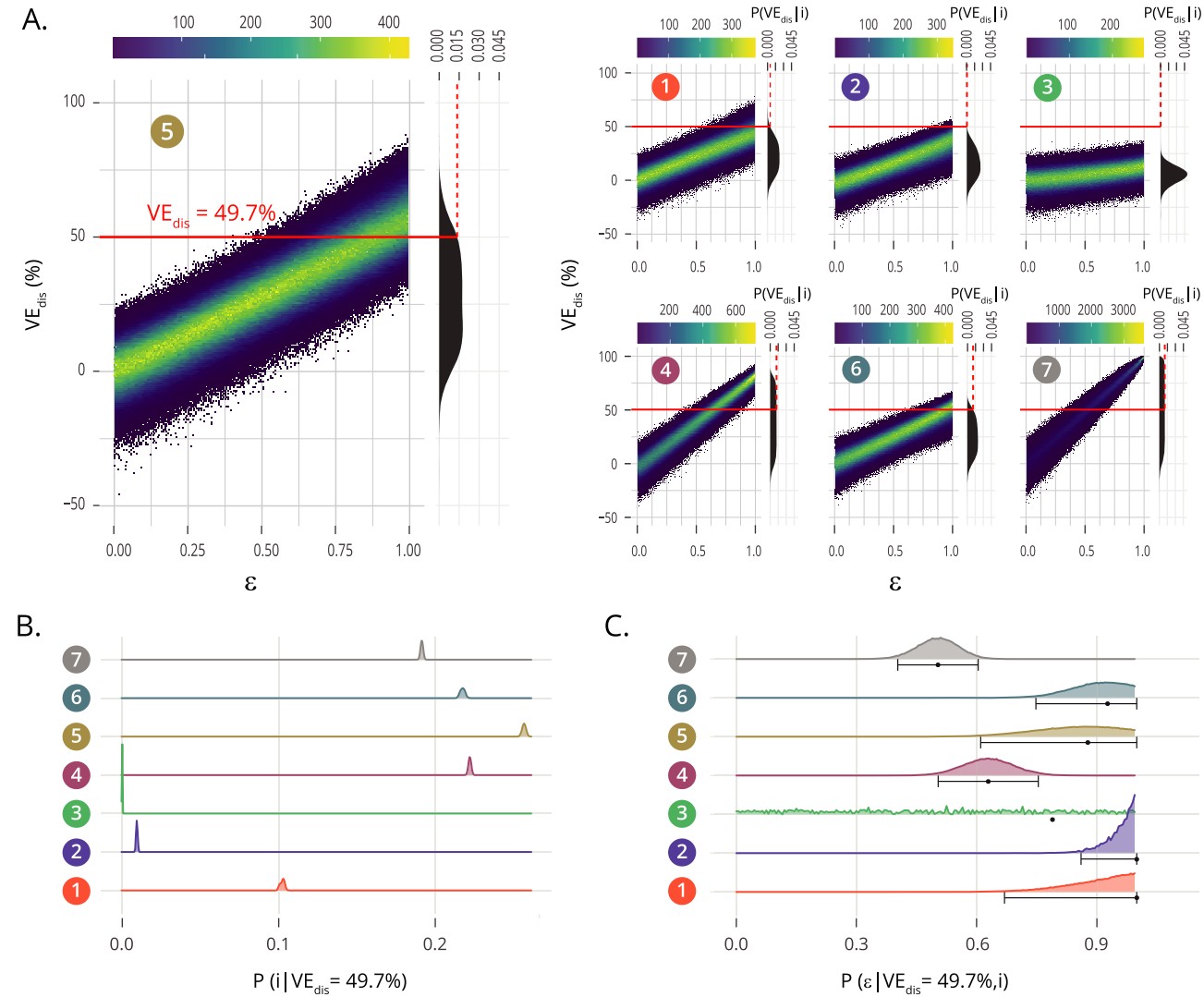

**Fig. 3 | Bayesian analysis of possible modeling architectures underlying a trial-derived observation of vaccine efficacy. A** Absolute frequency density distributions of efficacy values $VE_{dis}$ obtained in sets of $N = 2 \times 10^6$ clinical trial simulations per model, uniformly distributed across the intrinsic vaccine efficacy parameter $\varepsilon$ (efficacy resolution: 0.005, with 10,000 realizations for each value of $\varepsilon$). Red horizontal lines mark the PoD efficacy observed in the M72/AS01$_E$ trial $VE_{dis} = 49.7\%$. Along with each bi-dimensional density cloud, we represent its marginalized frequencies over the vertical axis, obtained upon adding simulation results over all possible values of $\varepsilon$ for each model. These density curves capture the marginalized likelihoods $P(VE_{dis}|i)$. Red dashed lines capture their value at the observed efficacy, that is $P(VE_{dis} = 49.7\%|i)$. **B** Marginal posteriors $P(i|VE_{dis} = 49.7\%)$, capturing the relative compatibility of each model with respect to the efficacy observed in the M72AS01$_E$ trial. **C** Distribution $P(\varepsilon|VE_{dis} = 49.7\%,i)$ of the intrinsic vaccine efficacy parameter $\varepsilon$ in each model type, given the observed efficacy $VE_{dis} = 49.7\%$, along with mean and 95% confidence intervals associated to them. For M3, the CI was omitted, for it spans the entire range $\varepsilon \in [0,1]$, as the model fails systematically to produce simulation instances compatible with the observed $VE_{dis} = 49.7\%$. Source data are provided as a Source Data file.

illustrate how it can be used to reduce arbitrariness from impact evaluations based on transmission models. A typical line of action for prospective impact evaluation of a vaccine consists of three steps: (1) implementing a transmission model accommodating a sensible vaccine description defined a priori. (2) Infer the vaccine parameter(s) conditional on the model structure that provide an optimal agreement with trial data, and (3) produce model-based forecasts of vaccine impact. A potential problem with this approach is, of course, that there exist many vaccine descriptions that can be adopted in the second step, and that they may lead to substantially different impact forecasts.

In order to illustrate this problem and quantify its importance, we capitalize on the same transmission model used above to infer forces of infection and fractions of fast vs. slow progressors[23]. This model was later adapted to allow for the description of the effects of the introduction of new vaccines[16]. Here, we have further adapted the model to

accommodate vaccine descriptions compatible with each of the seven models under analysis (see Methods & Supplementary text S1).

In Fig. 4A, we see the incidence reduction rate (IRR), evaluated in 2050, achieved by the introduction of a vaccine in 2025 on a vaccination campaign targeting adolescents (16–20 years old), under each of the seven types of models analyzed in this study in three high-burden countries: India, Indonesia, and Ethiopia. Here, the intrinsic efficacy modeled in each case corresponds to the expected value $\langle \varepsilon \rangle_i$ conditional to the model architecture and the vaccine trial PoD readout $VE_{dis}$ observed in the trial. In this exercise, vaccine coverage is ideally assumed to be 100%, and no efficacy waning has been modeled. In turn, the vaccine only protects IGRA+ subjects to avoid extrapolating its efficacy estimates to unexposed individuals, where efficacy evidence has not yet been gathered for this vaccine. According to each model description (see Fig. 2, and Supplementary Fig. S2), only PoD

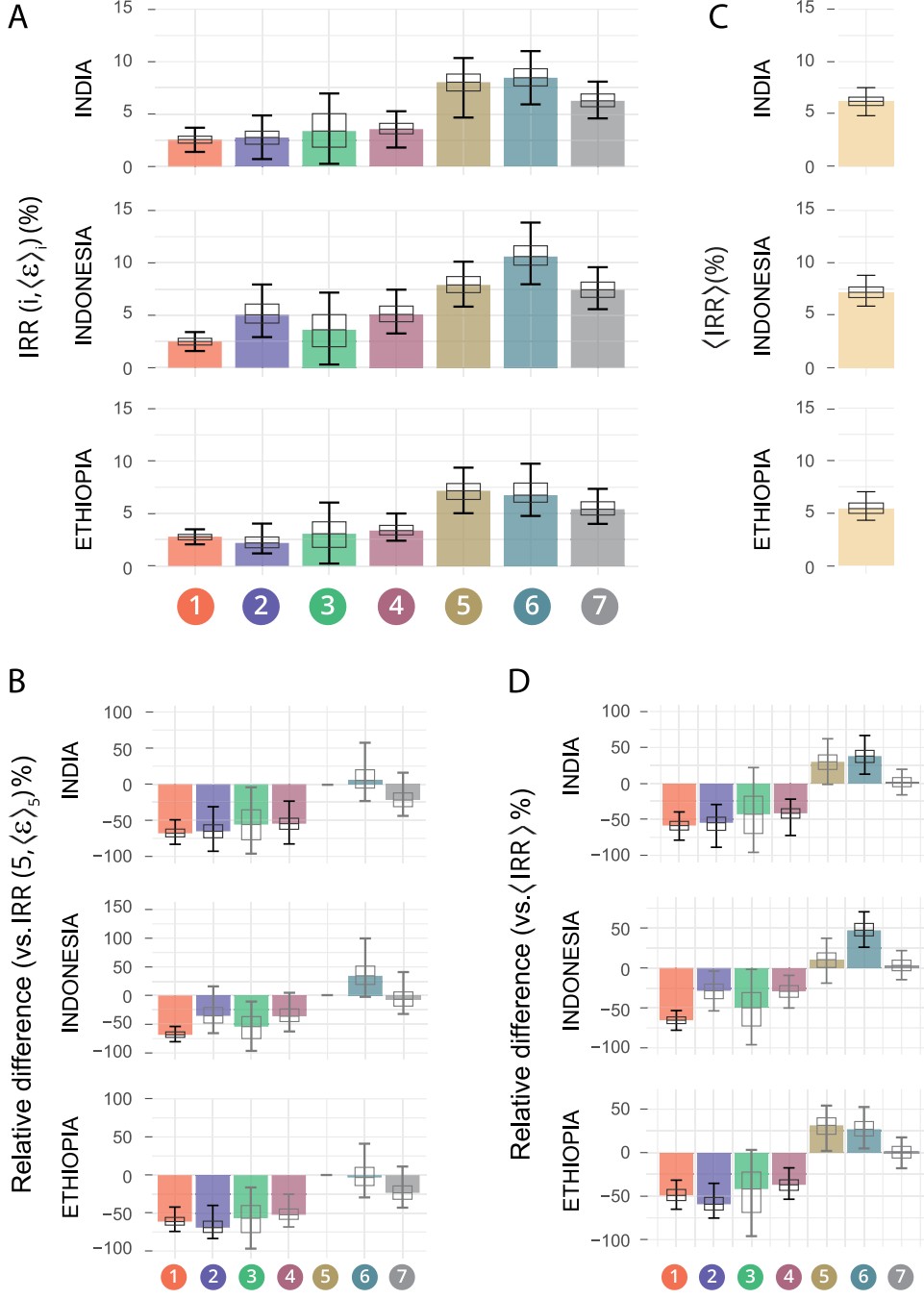

**Fig. 4 | Impact forecasts variation across model structures vs. mechanism-agnostic Bayesian estimates of impact. A** Vaccine impact forecasts obtained through the comprehensive transmission model introduced in ref. 16,23, when the vaccine is modeled according to each of the seven descriptions here discussed. **B** Relative differences between the impacts foreseen by each model and the model with maximum Bayesian posteriors (model 5). **C** Combined, mechanism-agnostic $\langle IRR \rangle$ estimates for the same impacts, in the same countries, where each of the seven models contributes proportionally to its Bayesian posteriors. **D** Relative differences between $\langle IRR \rangle$ and impacts foreseen by each individual model. In all panels, bars capture the median impact, boxes represent the inter-quartile range and error bars represent the 95% CI from sets of $N = 500$ impact simulations. *P* values are obtained as the fraction of simulations yielding impact estimates crossing zero, over a total set of $N = 500$ impact simulations (one-tailed empiric test). *P* values are further adjusted for multiple testing using Bonferroni correction with $N = 63$ tests. Black error bars correspond to significant statistics (Bonferroni-adjusted $p < 0.05$). Source data are provided as a Source Data file.

effects (no PoI or PoR) are included in our models. These impacts range from 2.2% of IRR in 2050 (95% CI: 1.2–4.0, as foreseen by model model 2 in Ethiopia), to 10.6% of IRR (95% CI: 8.0–13.8) foreseen by model 6 in Indonesia.

Then, we observed that many of the differences in vaccine impact that emanate from different vaccine models within the same country are statistically significant. In Fig. 4B we illustrate the relative

differences in IRR foreseen in each country by each vaccine model and the model with the highest posterior probability (model 5), describing a vaccine that protects against endogenous reactivation of LTBI and primary TB at once. These differences are statistically significant in 6 out of 18 cases (Bonferroni-adjusted *p* values < 0.05), and account for as much as 69.3% of the impact foreseen by model 5 in the most extreme case, $(IRR(2, \langle \varepsilon \rangle_2)$ impact lower than $IRR(5, \langle \varepsilon \rangle_5)$ in Ethiopia).

These results evidence the importance of removing arbitrariness from modeling choices of vaccine descriptions.

This can be achieved, at least within the family of models under analysis, using our Bayesian approach. In short, we propose considering Bayesian estimates of expected vaccine impact $\langle IRR \rangle$ as a mean of the impacts foreseen by each type of vaccine $IRR(i, \langle \varepsilon \rangle_i)$, weighted by the marginal posteriors $P(i|VE_{dis} = 49.7\%)$. The results of this exercise are presented in Fig. 4C for India, Indonesia and Ethiopia. In the impact forecasted in Fig. 4C, IRRs range from 13.35% in Ethiopia vs 1.99% in India, in line with results provided in other recent modeling studies for vaccines of comparable profiles, in comparable vaccination strategies[24,25]. As with comparisons across models, deviations of individual vaccine descriptions with respect to $\langle IRR \rangle$ range between +47.3% ($IRR(6, \langle \varepsilon \rangle_6)$ above $\langle IRR \rangle$ in Indonesia) and −65.4% ($IRR(1, \langle \varepsilon \rangle_1)$ below $\langle IRR \rangle$ in Indonesia), and are statistically significant in 9 over 21 occasions (Bonferroni-adjusted $p$ values < 0.05, Fig. 4D). This highlights again the risk of adopting a priori a given dynamical structure for vaccine descriptions in transmission models, and the convenience of adopting a Bayesian approach on this problem as we propose here.

## Discussion

In a disease with a complex transmission chain, such as TB, vaccine mechanisms can be modeled in many different ways, some of which can be rendered compatible with clinical trial observations and yet produce divergent results when plugged into transmission models for their prospective evaluation. To solve this problem, we propose a method that combines *in-silico* simulations with actual trial results to quantify the relative compatibility of different vaccine descriptions with trial-derived observations. These model-to-data compatibility metrics are nothing but Bayesian posteriors which we use as weights to retrieve expected vaccine impact forecasts where models that are more compatible with trial observations contribute more than those in conflict with data. By doing this, we provide a rationale that helps circumventing the need to make arbitrary modeling decisions with respect to vaccine mechanisms, which may bias their quantitative conclusions.

The discussion addressed here is pertinent within the context of TB vaccines development, since vaccines activating certain immune pathways and responses may exert different effects on the risk of developing TB associated to different routes to disease. As a case example of the potential of our approach, we analyzed the phase 2b efficacy trial of the promising vaccine candidate M72/AS01$_E$, conducted on individuals previously exposed to the pathogen (IGRA+). Here, we produced weighted averages for the impact of this vaccine, to conclude that M72/AS01$_E$ is expected to lead to an IRR of 6.22%, CI (4.85–7.52), 7.20% CI (5.88–8.82) and 5.44%, CI (4.30–7.02) evaluated in 2050 in India, Indonesia and Ethiopia, respectively, for a vaccine applied on adolescents starting on 2025, assuming perfect coverage and no efficacy waning, and assuming that previous exposure is needed for protection. These impacts are modest, implying that wider vaccination campaigns would be necessary to meet the End-TB strategy goals if the efficacy profile of this vaccine is consolidated in further, phase 3 studies and no better tool is at hand.

Using our method, we were able to assign different posterior probabilities to each of the seven vaccine models proposed. The magnitude of the posterior of each model depends on the marginalized likelihoods, integrated over all possible values of the vaccine efficacy parameter. These essentially depend on whether the maximum amount of TB cases preventable by each vaccine model is enough to explain the protection level observed in the trial, and, when we compare models with large enough maximum preventable fractions, on how frequently each model is able to generate simulated trials compatible with the observed vaccine efficacy. Our analyses showed that models 1, 2, and 3, each of which tackling a single route to disease, show lower posterior probabilities than models acting on

either two (models 4, 5, and 6) or all three routes to disease (model 7). Furthermore, the vaccine model offering the highest posterior probabilities given the trial result is model 5, where vaccine PoD leans on protection against endogenous reactivation of LTBI and primary TB, even though models 4, 6, and 7 show similarly high posterior probabilities. In this sense, the potential of our approach to disentangle specific vaccine mechanisms with better specificity than what is presented here could be further exploited if applied to the analysis of multi-centric trials conducted on sites with divergent TB burden distributions across age strata and routes to disease, and, unlike the example analyzed here, including participants distributed more homogenously across sites. In the case that we analyze here, the estimated distribution of TB cases across routes to disease is very similar in the three countries in the study (South Africa, Kenya, and Zambia, see Fig. 1), and a majority of trial participants come from the South Africa site (>80%), discouraging disaggregating the analysis per site. Should further efficacy data for this or other vaccines be collected, based on trials where a minimum number of participants per site and age strata is prioritized (additionally to prioritizing a minimum aggregated number of participants) this type of Bayesian approach could be stratified per site or age cohort, integrating more than just one efficacy observation. This, in turn, would unlock the estimation of more decisive Bayesian posteriors for the different vaccine models proposed.

In spite of these precautions, the entire set of model posteriors constitutes a meaningful resource that unlocks producing vaccine impact forecasts that are mechanism-agnostic. Using these posteriors as weights of the impact forecasts produced from each of the seven proposed models we obtain a Bayesian impact estimate that does not lean on any vaccine mechanism assumption. Importantly enough, our Bayesian estimates for the M72/AS01$_E$ vaccine impact are broadly compatible with those produced by model 7 alone, which is an architecture that has recently been used to produce the first impact forecasts for vaccines similar to M72/AS01E[24,25]. This suggests that the analyses presented in these references would not be incurring relevant bias in this particular case due to the implicit mechanistic assumptions made in their vaccine modeling choices. However, it is equally important to highlight that this does not guarantee that model 7 could be generally considered less prone to bias than other models, for the situation could be different for other vaccines, or even for this same vaccine after more evidence becomes available.

The approach here introduced fosters important limitations. On the one hand, the implementation of the clinical trials simulations requires estimating a series of epidemiological parameters a priori, including the fraction of individuals in the fast vs. slow progression reservoirs, rates of re-infection, and fast progression to disease; all conditioned (at least) by age stratum and epidemic setting in order to combine them, at a later stage, to describe the global study population. This was done in this study by adapting the coarse granularity of participants' age groups reported in the trial[9,10] (participant numbers were reported for three broad groups of 18–25, 25–30, and 30–50 years old) to the smaller age groups used in the model (seven 5 years-wide age strata: from 15–20 to 45–50) after assuming unbiased representation of the smaller age groups in each country in the wider cohorts reported in the trial. Similarly, we assume that the overall epidemic risk in the countries of the trial was representative of the overall situation in each specific setting in the year of the study. Admittedly, exact age distributions of the participants (whenever available, and possibly complemented with further information about risk factors), and more relevant information on incidence levels at the specific settings could be used in order to refine quantitative conclusions. Specifically, it could be thought that forces of infection used here are likely to underestimate the actual values observed in the trial, since trial settings are chosen by their

typically high transmission levels. If true, that may alter the relative weight of different routes to disease in our analysis, biasing our conclusions. Although it would be very valuable to count with empiric estimates for the force of infection in the trial sites, the estimates that we obtained, ranging between -4% and 11% depending on the country and the age group, are broadly compatible with expected values of the annual rate of infection in high TB burden settings, according to a recent study by Dowdy and Behr[26]. This study concludes that, unlike classical estimates for this parameter, adult populations in contemporary high-burden settings may present annual rates of infection between 5% and 10%, or even higher; a range that is compatible with our findings. Similarly, IGRA+ clinical trial designers should include strategies to explicitly quantify the fraction of the participants who underwent recent vs remote IGRA conversion before the beginning of the trial, which would remove the need to estimate the relative sizes of fast vs slow progression reservoir from transmission models. This could be done directly (i.e. by including an IGRA screening phase lasting circa one year before trial starts, where individuals who are initially testing negative are re-evaluated to capture a fraction of recent IGRA conversions before the beginning of the study), or, perhaps more feasibly, by using bio-markers of time since IGRA-conversion, a promising possibility that is technically available, as recently demonstrated in[27]. In summary, including protocols to produce empiric estimates of F vs. L relative weights in the trial sites, as well as forces of infection, instead of using model-based estimates as we do here, would be extremely helpful however difficult the logistics of the task may result in the practice.

On the other hand, it is important to highlight that our method, as implemented here, only permits vaccine descriptions where mechanisms are either absent, or present to the same extent, but does not accommodate more general situations where all vaccine mechanisms may be present with different intrinsic efficacies. Generalizing the formalism to deal with *leaky* vaccines -where different efficacy values are permitted, associated with different routes to disease, in the same model-, would unlock descriptions of more general vaccine behaviors. However, it is key to acknowledge that the amount and quality of efficacy data needed for generalizing our method in that direction is currently unavailable, for example, for the M72/AS01$_E$ vaccine case. Again, it would be necessary to count with enough participants distributed across locations in multi-centric studies, and/or age groups, where baseline distributions of estimated cases associated with each of the TB routes were divergent enough. Using that information, vaccine efficacy could be analyzed independently in different subgroups of data, producing more decisive posterior estimates of the mechanisms at place and their relative efficacy, also in a leaky vaccine scenario[18,19].

The method proposed in this study can be used for interpreting clinical trials for vaccine efficacy against active TB (PoD) conducted on IGRA+ individuals, and it can be extended to other trial designs, even for diseases obeying different transmission dynamics structures. For example, it could be extended to the study of trials conducted with IGRA- individuals, where PoD mediated by PoI would emerge as an additional vaccine mechanism to integrate within the framework. It can furthermore be used coupled with any transmission model of choice (see ref. 17 for an exhaustive review of most recent modeling tools described in recent literature for TB), as long as it accommodates the description of the different routes to disease and mechanisms of action here described.

By adapting our method to these situations, it will be possible to produce less arbitrary model-based impact forecasts based on vaccine descriptions where the knowledge about the vaccine behavior is incomplete.

## Methods
### Basal model calibration

The basic model describing TB acquisition of participants in the placebo arm during the trial (Fig. 1A), can be expressed through the following system of ordinary differential equations:

$$\dot{L} = -r_L L - \beta pqL$$
$$\dot{F} = -rF + \beta pqL \tag{1}$$
$$\dot{D} = rF + r_L L$$

where we consider an endogenous LTBI reactivation rate centered around $r_L = 7.5 \times 10^{-4}\, y^{-1}$ (95% CI $6.37 \times 10^{-4}$–$8.63 \times 10^{-4}$) whereas fast progression rate to TB is centered in $r = 0.9\, y^{-1}$ (95% C.I. 0.765–1.035). These values, widely adopted in the modeling literature[28–30], are in turn broadly compatible with empirical estimates (reviewed in ref. 22 -$r$-, and [31] -$r_L$-). According to ref. 32, we consider that LTBI individuals have a 79% less risk of progressing to TB upon re-infection, that is, $q = 0.21$ (95% C.I. 0.14–0.30). Finally, the probability of fast progression is centered around $p = 0.15$ (95% C.I. 0.10–0.20)[21,33,34]. With these parameters fixed (drawn in each realization from distributions compatible with expected values and C.I.s), the transmission rate $\beta$ is initially calibrated for each country, and within each age group (14 five years-wide age groups, from 0–5 to 65–70, plus a last age group gathering people above 70 years old: 15 groups in total). To do so, we resort to the same detailed transmission model used to evaluate vaccines impact (see sub-section on vaccine impacts below and Supplementary text S1)- bound to fit the incidence and mortality burden reported by the WHO between 2000 and 2018 (Fig. 1C, and Supplementary Fig. S2)[23]. The same model is used to produce estimates of the fraction of fast vs. slow progressors among IGRA+ populations in each country and age group (see Fig. 1D, E for the fitted values of $\beta$ and the fractions of slow vs fast progressors per country and age group).

Using these dynamical parameters, the dynamics in the placebo arm of the trial are simulated, according to the system of ODEs in Eq. (1), employing a version of the Gillespie algorithm where the reservoir F is mirrored in order to allow for independent tracking of the individuals undergoing primary TB who were initially in F as well as the individuals following the re-infection route to disease: L→F→D. As for individuals in the vaccine arms of the cohort, each all-or-nothing vaccine model can be mapped onto a combination of the epidemiological parameters ($r_L$, $r$, or the product $\beta pq$, see Supplementary text S1) being turned to zero for a fraction $\varepsilon$ of the vaccinated individuals. Through this approach we produce different parametrizations for the eventual prevention-of-disease (PoD) that is conferred by the vaccine under analysis. It is important to highlight that we are modeling only PoD vaccines whose action is not concomitant with neither prevention of infection (PoI), nor prevention of recurrence (PoR). Each in-silico trial comprehended the simulation of a fraction of individuals in different age groups, ranging from 18–50 years old, in three different African countries, with each country-age group combination being characterized by specific epidemiological parameters. We estimated these fractions from the reported participant distributions across age strata and country reported in ref. 9 (See Supplementary text S1 for details). Once the result of each simulation is obtained through the Gillespie algorithm, the PoD efficacy is estimated as 1 minus the ratio of cases observed in the placebo and intervention arms.

We simulated $N = 10,000$ trials for each model and value of the intrinsic efficacy; with 200 values of the intrinsic efficacy $\varepsilon$ uniformly distributed in the range [0,1]. Each one of these instances involves the simulation of the dynamics in both cohorts in three countries and within seven age groups (16–20 to 46–50 years old, age group width = 5 years), that are combined into a single efficacy readout per instance. This yields a total number of trials simulated equal to $4.2 \times 10^7$

per model, (200 intrinsic efficacies × 10,000 instances × 7 age groups × 3 countries); that is, $1.47 \times 10^8$ trials simulated in total for the seven models analyzed.

## Bayesian analyses

Let us consider the likelihood $P(VE_{dis} = 49.7\%|i,\varepsilon)$ that each of the possible models defined by the combination of parameters $\{i,\varepsilon\}$, (where the integer index $i \in \{1,2,\ldots,7\}$ determines the specific vaccine mechanism(s) at work (i.e., the vaccine model type), and the continuous parameter $\varepsilon \in [0,1]$ captures its intrinsic efficacy) generates a PoD efficacy estimate compatible with the one observed for M72/AS01$_E$. Using this likelihood term, we apply Bayes rule to define the posterior probability associated with each particular model:

$$P(i,\varepsilon|VE_{dis} = 49.7\%) = \frac{P(VE_{dis} = 49.7\%|i,\varepsilon) \cdot P(i,\varepsilon)}{\sum_{i'} \int_{\varepsilon'} (VE_{dis} = 49.7\%|i',\varepsilon') \cdot P(i',\varepsilon') \cdot d\varepsilon'} \quad (2)$$

And derive a model-type posterior probability, by integrating over all possible intrinsic efficacy values as follows:

$$P(i|VE_{dis} = 49.7\%) = \int_{\varepsilon} P(i,\varepsilon|VE_{dis} = 49.7\%) \cdot$$
$$d\varepsilon = \frac{\int_{\varepsilon} P(VE_{dis} = 49.7\%|i,\varepsilon) \cdot P(i,\varepsilon) \cdot d\varepsilon}{\sum_{i'} \int_{\varepsilon'} (VE_{dis} = 49.7\%|i',\varepsilon') \cdot P(i',\varepsilon') \cdot d\varepsilon'} \quad (3)$$

If we consider uniform non-informative priors in Eq. (3) (that is $P(i,\varepsilon) = P(i',\varepsilon') \forall (i, i', \varepsilon, \varepsilon')$), the model-type posterior can be obtained from the marginalized likelihoods:

$$P(VE_{dis} = 49.7\%|i) = \int_{\varepsilon} P(VE_{dis} = 49.7\%|i,\varepsilon) \cdot P(i,\varepsilon) \cdot d\varepsilon \quad (4)$$

Which we estimate from the density distributions of the PoD efficacy readouts $VE_{dis}$ obtained from each model using the value of Kernel density estimators (R package KerSmooth) of the frequency of trials evaluated at $VE_{dis} = 49.7\%$. Plugging the numerical estimates of $P(VE_{dis} = 49.7\%|i)$ into Eq. (3) allow us to quantify the relative support in the data for the seven different vaccine descriptions provided. Confidence intervals for these model posterior estimates represented in Fig. 3B are obtained by bootstrapping the calculations $N = 5000$ times, each of which is obtained by sampling with replacement $N = 1,000,000$ trial simulations.

Then, we also estimate the intrinsic efficacy values $\varepsilon$ that are most compatible with the given observed efficacy against-disease readout $VE_{dis} = 49.7\%$ under each of the model type descriptions, this time applying the Bayes rule over each model type independently:

$$P(\varepsilon|VE_{dis} = 49.7\%, i) = \frac{P(VE_{dis} = 49.7\%|i,\varepsilon) \cdot P(i,\varepsilon)}{\int_{\varepsilon'} P(VE_{dis} = 49.7\%|i,\varepsilon') \cdot P(i,\varepsilon') \cdot d\varepsilon'}$$

Where likelihood terms $P(VE_{dis} = 49.7\%|i,\varepsilon)$ are estimated from the simulations using Kernel density estimates obtained for each of the $N = 200$ values of $\varepsilon$ covered. The first momentum of these posterior distributions corresponds to the expected values of the intrinsic efficacy parameter under each model type, that is:

$$\langle\varepsilon\rangle_i = \int_{\varepsilon} P(\varepsilon|VE_{dis} = 49.7\%, i) \cdot \varepsilon \cdot d\varepsilon = \frac{\int_{\varepsilon} P(VE_{dis} = 49.7\%|i,\varepsilon) \cdot P(i,\varepsilon) \cdot \varepsilon \cdot d\varepsilon}{\int_{\varepsilon'} P(VE_{dis} = 49.7\%|i,\varepsilon') \cdot P(i,\varepsilon') \cdot d\varepsilon'}$$

which is captured in Fig. 3C, along with its confidence intervals obtained from the posterior distribution $P(\varepsilon|VE_{dis} = 49.7\%, i)$ itself, fitted to a normal distribution.

Finally, we build model-based Bayesian estimates of vaccine impact as a weighted linear combination of the impacts foreseen by each type of vaccine, expressed as incidence reduction rates, as follows:

$$\langle IRR\rangle = \sum_{i} P(i,|,VE_{dis} = 49.7\%) \cdot IRR(i,\langle\varepsilon\rangle_i)$$

where the seven incidence reduction rates $IRR(i,\langle\varepsilon\rangle_i)$ are computed using the same comprehensive transmission model used to estimate transmission rates $\beta$ and fractions of prevalent fast vs slow progressors[23].

## Impact evaluations

In this study, we make use of the transmission model introduced in ref. 23, and generalized to describe the introduction of novel vaccines in[16]. This model constitutes a conceptual extension of the basic model sketched in Fig. 1A, where different types of disease are considered, along with treatment outcome dynamics and eventual relapses. The model integrates demographic data and empiric mixing patterns among age strata along with epidemiological parameters and TB burden trends (incidence and mortality), in order to produce baseline incidence and mortality forecasts per country, as well as vaccine impact evaluations. For further details on the model, the reader is referred to the Supplementary text S1, along with references[16,23].

## Reporting summary

Further information on research design is available in the Nature Portfolio Reporting Summary linked to this article.

# Data availability

All data supporting the findings described in this manuscript are available in the article and in the Supplementary text S1 and Source Data, and from the corresponding author upon request. Data concerning TB burden, as well as countries demographic structures is available in open databases[35,36]. The data concerning the M72/AS01$_E$ vaccine trial used here is publicly available at the original source[9,10]. Source data are provided in this paper.

# Code availability

Code with the implementation of the novel methods introduced in this study is available at GitHub (https://github.com/MarioTovarCalonge/Bayesian_Framework_TB_Vaccines) and at Zenodo (https://zenodo.org/badge/latestdoi/596053638)[37]. Those codes include algorithms written in C language (Gillepie algorithm-based implementation of clinical trial simulations), and in R (tested in version 3.6.3), with dependences, at different stages, to the following R packages: fanplot (v4.0.0), ggplot2 (v3.4.0), gridExtra (v2.3), kdensity (v1.1.0), KernSmooth (v2.23-20), viridis (v0.6.2), truncnorm (v1.0-8), minpack.lm (v1.2–2), nlsr (v2019.9.7), iterators (v1.0.14), for each (v1.5.2), doParallel (v1.0.17), and dplyr (v1.0.10).

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

## Acknowledgements

We thank M. Gutiérrez for the graphic design assistance. Funding Government of Aragón: Ph.D. contract order IIU/796/2019 (M.T.), grant E36-23R (FENOL: M.T., Y.M.), and grant B49_23R (NeuroBioSys: J.S.). MCIN/AEI/ 10.13039/501100011033 and "ESF Investing in your future": grants PID2019-106859GA-I00 and RYC-2017-23560 (J.S.), and grant PID2020-115800GB-I00 (Y.M.), Banco Santander: Santander-UZ 2020/0274 (Y.M.). Soremartec S.A. and Soremartec Italia, Ferrero Group. (Y.M.). The funders had no role in study design, data collection, and analysis, decision to publish, or preparation of the manuscript.

## Author contributions

M.T., J.S., and Y.M. designed research; M.T. performed research; M.T. and J.S. analyzed data, and discussed results with Y.M. J.S. and M.T. wrote the article with inputs from Y.M. All authors read and approved the final version of the manuscript.

## Competing interests

The authors declare no competing interests.
