## [Peer review file · Nature Communications]

REVIEWER COMMENTS

Reviewer #1 (Remarks to the Author):

Brief summary of paper

This is a very interesting and potentially important manuscript proposing a mathematical and statistical framework that could be used to estimate the possible ways a vaccine could prevent disease (or other health outcome). It is applied to Tuberculosis, which is a great case study, as a trial disease outcome could be prevented via at least 3 routes.

The framework is presented, and they make some potentially useful statements relevant for TB trial design and interpretation.

Summary of opinion

The aim is excellent, and the methods look sound in general, but the paper needs a bit of work, primarily on making methods and results clearer, to maximise its use for TB trial design and interpretation.

Data and methods

Major

Line 116. "To gather estimates of these key parameters, we calibrate an exhaustive TB infection transmission model previously developed [23] in each of the countries involved in the study, from where we retrieve estimates of the epidemiological parameters that play a role in the minimal model of figure 1A, as well as the relative prevalence of fast (F) vs slow (S) individuals in the population corresponding to 2015, the year that the M72/AS01E study took place (see Methods and Supplementary text S1)." Needs much more explanation in the main text as to how these models were created, and showing the calibration to the data to show that the projections are plausible? Why should I, as a reader, believe that there is virtually no primary TB in the Zambia population? But lots in SA? Is this difference credible? (on the face of it, not very). What in the data or modelling explains this prediction? The reader needs to believe this before taking any of the subsequent results seriously. >> Add this explanation

Somewhere in methods, need to be clear somewhere in methods that the authors are only modelling POD action, (ie not POI), and in the discussion, that actually, the impact of POD vaccines could be explained entirely from a POI effect). The authors can explain that this vaccine action scenario is not within the scope of this analysis. Also, it would be interesting to know whether this question could be explored using this framework in the discussion. >> Add description

The authors make the strong assumption that vaccines act as 'all-or-nothing', rather than 'degree'. When work has shown that all-or-nothing tends to overestimate vaccine impact vs degree protection. Assuming degree protection therefore may affect the results. I would expect to see this mentioned in the discussion, and ideally a comment on what the likely effect might be (is possible to speculate). Given vaccines are quite likely to work in this way, how much can we use the results? >> Add to the discussion

Before we read the results we need to know the assumed vaccine characteristics and how it is targeted. At the moment we only find out that the vaccine is deployed routinely to 16 year olds and initially mass to 17-20 year olds in the supporting material. And I am not sure if we are ever told the duration of efficacy. Is it pre or post-infection status required for efficacy? Does it have POI efficacy (I

don't think so)? What is assumed coverage? All this needs to be said upfront, alongside the vaccine structure scenarios, so results can be interpreted.

>> Add description

L 335 "bound to fit the incidence and mortality burden reported by the WHO between 2000 and 2019 [23]. " Need to show model fits to these data (ie incidence and mortality over time showing model and data) to help the reader understand if models are a good representation of setting. Can be in supporting material.

>> Add description

Supporting material. Model description is to another paper. The paper needs to include a complete description of the model that was used, so the work can be reproduced. Referring to other papers, when there are very likely to be differences does not allow reproducibility.

>> Add model descriptions/equations

Minor

L 328 "where we consider an endogenous LTBI reactivation rate centered around $r_L = 7.5 \cdot 10^{-4} y^{-1}$ (C.I. $6.37 \cdot 10^{-4} - 8.63 \cdot 10^{-4}$) whereas fast progression rate to TB is centered in $r = 0.9 y^{-1}$ (C.I. $0.765 - 1.035$) [33]. "

>> This refers to a modelling paper. Would be better to refer to the primarily empirical lit.

L 333: "and within each age group, "

>> Please state age groups here.

Interpretation of findings

Major

The abstract needs more details on the actual findings from the TB case study. At the moment the abstract primarily describes what will be done, but not what was found, or what it means

>> It would be good to add a key finding for the TB case study? What are the implications: for whom?

Line 173 ".the observed PoD efficacy readout reported for the vaccine M72/AS01E [10] appears more compatible with models 4, 5, or 6, each ..."

>> Not enough description of this fig. Need to justify statement by describing what reader can see in fig

Line 176 "is not the model where the vaccine is assumed to interrupt equally the three process, " This is really important - The authors need to explain **why this is the case** ie explain the exact reason the model is predicting this. This is not a mystery - they have the model that they have created that can tell them the underlying reason exactly why these vaccine effect models are preferred over the others. This is essential so reader can assess the plausibility of this critical finding - the result could just be an artefact of how they put the model together, or a quirk of the data.

>> Add explanation

L269 "Using our method, we found that the vaccine model offering the highest posterior probabilities given the trial results is model 6, where the modelled vaccine PoD acts on protection against endogenous reactivation of LTBI and TB upon reinfection. This result is interesting, for it suggests that vaccine effects on halting on-going fast transition to TB on individuals infected not long before vaccination (the only mechanism absent from model 6) are not more important, or likely to be present in this particular vaccine with the data at hand, than the other two". Ditto the previous comment. **What** is it about the created model scenarios and the vaccine structures that have led to this result? This is crucial for understanding if the results should be seriously or not (and, for example, other modelling papers should use the posterior values in their modelling work).

>> Add explanation

L215 "These results evidence the importance of removing arbitrariness from modelling choices of vaccine descriptions". This is undoubtedly true. but until the authors explain mechanistically **why** the model scenarios and the data have led to results we have seen, this paper has yet to provide good evidence that the "arbitrariness has been reduced". As elsewhere, authors need to explain mechanistically why the confrontation between model and data has led to structures 5 and 6 being preferred over the other structures. If plausible, this may lead reader to believe that plausible evidence has been created (and the results are not just model artefact/quirk)

>> Add explanation

Para L217. Interestingly, I notice that the simpler vaccine protection structure represented in structure '7' (vaccine prevents all routes equally) is the closest of all the 7 structures to the results from this complicated analysis, two-thirds of the time (ie in India and Indonesia it ranks first, and it ranks third in Ethiopia) (Fig 4D). As such the authors should really say, that if a simpler representation of vaccine effect was needed (because, eg, of computational resources or time limitations or unavailability of code for this new method), using structure 7 may be a good choice. Further, I notice that the authors say that structure 7 is the structure used in the modelling papers highlighted by the authors (refs 25 and 26). As such, this very elegant new work seems to suggest that the results from the simpler work (25 & 26) may be more robust than other papers to this potential source of bias.

>> It would be good to make these points in the manuscript

L 281 "the potential of our approach to disentangle specific vaccine mechanisms with better specificity than what is presented here could be further exploited if applied to the analysis of multi-centric RCT with divergent TB demographics and, unlike the example analyzed here, homogeneous representation across sites"

>> This is really important but not clear enough yet to be useful. The authors need to say what they mean by "divergent TB demographics" and "homogeneous representation across sites". What exactly should a trialist do to provide the information needed?

L 293 "This was done in this study by assuming unbiased representation of the age groups in each country in the study cohorts, and assuming that the overall epidemic risk in the countries of the trial were representative of the overall situation in each specific setting in the year of the study". >> This assumption is plainly not true as the trial populations were selected to be higher risk than general populations, to decrease sample size.

Can the authors speculate here on the likely bias this might introduce to their findings and conclusions?

Minor

L186 "These expected efficacy estimates illustrate a sensible feature of our model approach, namely, that a given PoD readout $[VE]_{dis}$ must be mapped to lower intrinsic efficacy ϵ values when the vaccine is able to halt progression to disease through all possible routes than when it acts on a subset of them"

>> This could have been said before any modelling is done. I suggest it is phrased as a "As expected,"

L 310 "especially for the analysis of balanced multi-centric trials "

>> What does 'balanced' mean here?

L241 Discussion. This para is really interesting, but not essential to the value of the work, so could be cut/reduced to make room for the additional material I have suggested (if there is a word count issue)

Clarity

Major

Line 23 "unbiased framework ". This is a big claim, and not at all proven
>> Suggest change to "potentially less biased"

Minor

Line 53 - BGC revax not included in list of vaccines here, but is in the able
>> Add bcg revax to text

Line 53 "Remarkable" odd word choice as some of these design choices were very sensible eg testing bcg revax in IGRA-s (as thought only to work in IGRA-s)
>> Drop word?

Line 89 "Applying our Bayesian formalism, we show that not all possible vaccine mechanisms that a priori could be included in All-or-Nothing compartmental models -arguably the most widely used type of vaccine models used in the modeling literature [17-19]- are equally backed-up by this specific trial readout"
>> This is a very complicated sentence. Most of the middle can be cut

Line 91 "... -arguably the most widely used type of vaccine models used in the modeling literature [17-19]..."
>> If this is not dropped, it needs to be justified - state how many of the TB vx papers make this assumption, and how many do not. (the review says there are about 30, so it is not a big task to find out)

Line 106 "Vinnickys " Incorrectly spelt

Line 176 "in spite of this model architecture being the one used in most recent modeling works aimed at producing impact estimates for TB vaccines similar to M72/AS01E [25,26].
>> Authors need to justify this. How many of the models in the lit use the various mechanisms the authors describe? Again the lit is small and has been reviewed, so easy to come up with this data to support the statement.

Motivation/research question

Major

Given the work is presenting a framework that could be used by others, I would expect the model code with documentation to be available.
>> Share and document on (eg github)

Lit review/framing

Major

Line 73-6. "Yet, a majority of the models in the current literature on TB vaccine modeling base their forecasts on assuming that efficacy readouts unequivocally map onto specific combinations of action mechanisms without providing a plausible justification of this important modeling choices [17]." This is true, but this review paper cited does not make this statement.
>> Drop sentence, or if the authors want to refrain it, this statement needs to be justified. Eg a summary of which papers were in the 'majority' and which were not.

Reviewer #2 (Remarks to the Author):

This interesting manuscript takes a Bayesian modelling approach to the evaluation of TB vaccine efficacy, using data generated from the M72 vaccine efficacy trial. The authors are correct that although we can determine if a vaccine is effective in an efficacy trial, we cannot determine easily by which mechanism such efficacy is achieved. The authors suggest reasons why such an understanding might be important, which relate to real world efficacy in preventing transmission and real world effectiveness.

The paper is generally well written and explains the concepts clearly. The assumptions – ie that the efficacy observed with M72 / AS01e, may be because of 3 possible scenarios in each latently infected subject enrolled – are correct.

This modelling leads to some very interesting conclusions. E.g. the vaccine model offering the highest posterior probabilities is model 6, where vaccine PoD leans on protection against endogenous reactivation of LTBI and TB upon reinfection – I'm not quite sure how the authors extrapolate from this to conclude that the trial result was unlikely to be due to an effect of the adjuvant alone.

The 3 countries chosen to illustrate the point have different rates of LTBI in their populations – was this factored into the model in any way?

The limitations section contains some logistically implausible suggestions – e.g. that a year prior to enrolment subjects are recruited so the stability of their latent Mtb infection is determined. Whilst scientifically I can see this would be interesting, the logistics and cost would preclude this happening. Unfortunately our ability to accurately distinguish between long term and recent infection is far from established at the moment.

REVIEWER COMMENTS

Reviewer #1 (Remarks to the Author):

Brief summary of paper

This is a very interesting and potentially important manuscript proposing a mathematical and statistical framework that could be used to estimate the possible ways a vaccine could prevent disease (or other health outcome). It is applied to Tuberculosis, which is a great case study, as a trial disease outcome could be prevented via at least 3 routes.

The framework is presented, and they make some potentially useful statements relevant for TB trial design and interpretation.

Summary of opinion

1. The aim is excellent, and the methods look sound in general, but the paper needs a bit of work, primarily on making methods and results clearer, to maximise its use for TB trial design and interpretation.

We would like to thank the reviewer for their generally positive opinion of our work, and its potential relevance in the field of clinical trials design.

After reading carefully all their suggestions, we think that these pointed towards aspects of our initial submission that were either problematic or insufficiently justified and contextualized, and thus required substantial revision. Some of them have helped us to identify previously unnoticed issues in our analysis pipeline too, and to correct them. In the following lines we describe all the changes introduced in the main text (highlighted using change tracker) in the main text document, point by point. Importantly, **we believe that we have accommodated all reviewer's suggestions, without exception.**

We believe that the result is a significantly improved manuscript, for which we would also like to sincerely thank the reviewer for his/her thorough and extremely valuable report.

Data and methods

Major

2. Line 116. "To gather estimates of these key parameters, we calibrate an exhaustive TB infection transmission model previously developed [23] in each of the countries involved in the study, from where we retrieve estimates of the epidemiological parameters that play a role in the minimal model of figure 1A, as well as the relative prevalence of fast (F) vs slow (S) individuals in the population corresponding to 2015, the year that the M72/AS01E study took place (see Methods and Supplementary text S1)."

Needs much more explanation in the main text as to how these models were created, and showing the calibration to the data to show that the projections are plausible?

We agree. We now add new materials to ensure that the method followed is fully understood and can be reproduced:

In the main text (lines 172-190 in the document with highlighted changes: all references to main text lines will refer to that document in the following) we extend the explanation on how we use the exhaustive transmission model from ref. 23 to gather the missing inputs necessary to execute our clinical trials simulations: the force of infection and the $F/(F+L)$ ratio per country and age group.

the model structure, as well as the results of TB incidence calibration, and the estimates of the forces of infection and $F/(F+L)$ fractions are now included as additional panels in figure 1 (panels 1B-1E) for the three countries of the study. Additionally, we provide an extended version of the calibration data including also mortality trends as well as the three countries where the vaccine impact is extrapolated (India, Indonesia and Ethiopia) in the supplementary figure S1)

We have added additional text in the supplementary appendix detailing the technical specs of the model, and uploaded its entire code to Github. These extra details include the model equations, a list with epidemiological parameters used, as well as other data sources that the model integrates.

3. Why should I, as a reader, believe that there is virtually no primary TB in the Zambia population? But lots in SA? Is this difference credible? (on the face of it, not very). What in the data or modelling explains this prediction? The reader needs to believe this before taking any of the subsequent results seriously. >> Add this explanation

We do thank the reviewer for raising this important concern, since, in fact, Zambia's outlier profile in the distribution of events shown in figure 1 was a consequence of a code error that had passed unnoticed in our first submission, which in turn impacted downstream analyses. Thanks to the revision of this point required by the reviewer, we could identify and correct this issue, repeating, consequently, all the analyses in the article.

The result, as foreseen by the reviewer, outlines a distribution of TB cases across disease routes that is fairly similar in Zambia to what is observed in the other two countries.

4. Somewhere in methods, need to be clear somewhere in methods that the authors are only modelling POD action, (ie not POI), and in the discussion, that actually, the impact of POD vaccines could be explained entirely from a POI effect). The authors can explain that this vaccine action scenario is not within the scope of this analysis. Also, it would be interesting to know whether this question could be explored using this framework in the discussion. >> Add description

We agree with the reviewer in this appreciation, and we now added the mention to the fact that we are only modeling POD action, (methods section lines 695-698) as well as a brief comment in the discussion (lines 641-643) on how our approach could be extended to model vaccines providing POI as well.

5. The authors make the strong assumption that vaccines act as 'all-or-nothing', rather than 'degree'. When work has shown that all-or-nothing tends to overestimate vaccine impact vs degree protection. Assuming degree protection therefore may affect the results. I would expect to see this mentioned in the discussion, and ideally a comment on what the likely effect might be (is possible to speculate). Given vaccines are quite likely to work in this way, how much can we use the results? >> Add to the discussion

Following reviewer's suggestion, we have added a discussion of the implications of this modelling decision in the discussion (lines 629-637), where we also comment on how our

approach could be coupled with leaky-vaccine (also referred to as degree-vaccine) descriptions too.

6. Before we read the results we need to know the assumed vaccine characteristics and how it is targeted. At the moment we only find out that the vaccine is deployed routinely to 16 year olds and initially mass to 17-20 year olds in the supporting material. And I am not sure if we are ever told the duration of efficacy. Is it pre or post-infection status required for efficacy? Does it have POI efficacy (I don't think so)? What is assumed coverage? All this needs to be said upfront, alongside the vaccine structure scenarios, so results can be interpreted. >> Add description

We agree with the reviewer, and this is now detailed in the main text in the results section (lines 366-370).

7. L 335 “ bound to fit the incidence and mortality burden reported by the WHO between 2000 and 2019 [23]. ” Need to show model fits to these data (ie incidence and mortality over time showing model and data) to help the reader understand if models are a good representation of setting. Can be in supporting material. >> Add description

We agree, and this is now added in figure 1C (incidence data only), as well as in the supplementary appendix (figure S1). In the SI appendix we have included the incidence and mortality fits of the spreading model in all 6 countries involved in the work (the three countries where the M72/AS01E trial took place -South Africa, Kenya and Zambia-, as well as the three countries where the vaccine impact is then extrapolated to -India, Indonesia and Ethiopia-).

8. Supporting material. Model description is to another paper. The paper needs to include a complete description of the model that was used, so the work can be reproduced. Referring to other papers, when there are very likely to be differences does not allow reproducibility. >> Add model descriptions/equations

As suggested, we have now included the model equations and the parametrization in the supplementary materials (SI appendix). We have also uploaded the spreading model code to Github to ensure reproducibility, see https://github.com/MarioTovarCalonge/Bayesian_Framework_TB_Vaccines

Minor

9. L 328 “where we consider an endogenous LTBI reactivation rate centered around $r_L=7.5 \cdot 10^{-4} \gamma^{-1}$ (C.I. $6.37 \cdot 10^{-4}$ - $8.63 \cdot 10^{-4}$) whereas fast progression rate to TB is centered in $r=0.9 \gamma^{-1}$ (C.I. 0.765-1.035) [33]. ” >> This refers to a modelling paper. Would be better to refer to the primarily empirical lit.

We agree with the reviewer, and have modified the text accordingly, adding references to reviews of empiric evidence behind these parameter estimates as requested (see lines 659-661).

10. L 333: “and within each age group, ” >> Please state age groups here.

This is now described in the main text in lines 680-681.

Interpretation of findings

Major

11. The abstract needs more details on the actual findings from the TB case study. At the moment the abstract primarily describes what will be done, but not what was found, or what it means. >> It would be good to add a key finding for the TB case study? What are the implications: for whom?

We agree, and we have rewritten the abstract accordingly to include a description of the results found, as well as to reduce its word count according to the journal policy.

12. Line 173 “..the observed PoD efficacy readout reported for the vaccine M72/AS01E [10] appears more compatible with models 4, 5, or 6, each ...” >> Not enough description of this fig. Need to justify statement by describing what reader can see in fig

13. Line 176 “is not the model where the vaccine is assumed to interrupt equally the three process, ” This is really important - The authors need to explain ****why this is the case**** ie explain the exact reason the model is predicting this. This is not a mystery - they have the model that they have created that can tell them the underlying reason exactly why these vaccine effect models are preferred over the others. This is essential so reader can assess the plausibility of this critical finding - the result could just be an artefact of how they put the model together, or a quirk of the data.

I will answer these two comments together, for they are extremely related. We think this question is very important as well, and thank the reviewer for pointing that this aspect required further clarification.

To clarify this question, we have included an explanation of the patterns shown in figure 3B (lines 280-334), and we have included marginalized likelihood curves in figure 3A, (which are extremely instrumental to understand this), as well as a new panel in figure 2 (figure 2B, featuring maximum fractions of preventable cases by each vaccine), both of which serve to illustrate how the model posteriors depend on the observed value for VE_{dis} in the trial, relative to the maximum preventable fraction of cases predicted for each model.

As it is now written in the text: “The reason behind this emerging hierarchy between vaccine models is the relation between the observed VE_{dis} and the maximum fraction of events that are preventable by each type of vaccine (shown in figure 2B). Posteriors of models whose maximum fraction of preventable cases is smaller than the observed VE_{dis} are in turn smaller, meaning that the protection mechanisms present in these vaccines are likely insufficient to explain the observed trial result (models 1,2 and 3: see figures 2B, and 3A). In what regards the remaining vaccine models (4 to 7), all of them provide protection against several routes to disease, featuring maximum fractions of preventable cases that are well above the VE_{dis} value observed in the trial (see figure 2B). Their different posteriors can then be understood by comparing the relative frequency at which each model is expected to generate simulated values for VE_{dis} that are compatible with the trial observation, when all simulations at all possible values of ε , distributed around a uniform, non-informative prior, are considered (marginal density curves over the VE_{dis} axis in figure 3). Observing those curves, -which are simply the marginalized likelihoods, integrated over ε for each model as defined in Equation 4 (see Methods)- we see that models 4-6 show marginalized densities with a tighter spread around intermediate values of VE_{dis} than M7, thus showing higher values around the observed $VE_{dis} = 49.7\%$, and therefore, higher model posteriors than M7. Maximum fractions of preventable cases for models 4, 5 and 6 are smaller than that of model 7, which is equal to 1, since model 7 can potentially prevent all TB cases by blocking all routes to disease alike. This translates into a cloud of simulated data with a steeper slope in figure 3A, which in turn causes the flatter

marginal density curve for M7 than for models 4-6. These results, taken together, lead to the slightly lower posterior observed for model 7 than for models 4-6, evaluated at $VE_{dis} = 49.7\%$."

14. L269 "Using our method, we found that the vaccine model offering the highest posterior probabilities given the trial results is model 6, where the modelled vaccine PoD acts on protection against endogenous reactivation of LTBI and TB upon reinfection. This result is interesting, for it suggests that vaccine effects on halting on-going fast transition to TB on individuals infected not long before vaccination (the only mechanism absent from model 6) are not more important, or likely to be present in this particular vaccine with the data at hand, than the other two". Ditto the previous comment. ****What**** is it about the created model scenarios and the vaccine structures that have led to this result? This is crucial for understanding if the results should be seriously or not (and, for example, other modelling papers should use the posterior values in their modelling work). >> Add explanation

We need to comment several aspects about the fragment mentioned by the reviewer.

- First, we need to highlight that this result has changed upon the correction completed for this submission (see answer to question # 3 above), and the model that appears now with the largest posterior is model 5, instead of model 6. The reason for this shift in the top-ranked model -the rest of the rank remains essentially identical to what was found in the preliminary version of the analyses can be contextualized if we compare the marginalized likelihoods obtained for models 5 and 6 from the data in the first submission, and the one submitted now after the corrections:

Looking to these likelihood values, we first see a small variation in the maximum fraction of cases preventable by each vaccine, (red dashed lines) which is now slightly larger for model 5, and slightly lower for model 6. Those changes, however small, are the responsible of the slight shift

in the likelihood curves that is observed when comparing the previous submission and the corrected version of the analyses. Ultimately, these small shifts change appreciably the value of the likelihood evaluated at $VE_{dis} = 49.7\%$ (blue arrows), and finally, the marginal posteriors in figure 3B.

Beyond this important correction, we should admit two aspects highlighted by the reviewer:

- On the one hand, we agree with the reviewer in that it is important to explain what is the cause after which some model (or models) obtains higher posteriors than others in our analysis. To do this, we have added text in lines 280-334 (see question above) as well as in lines 435-525 here. Essentially, as we now write in the main text: “The magnitude of the posteriors of each model depends on the marginalized likelihoods measuring how likely it is for each model type to generate stochastic simulations that are compatible with the trial observation. This essentially depends on whether the maximum amount of TB cases preventable by each vaccine model is enough to explain the protection level observed in the trial, and, to a lesser extent, when we compare models with large enough maximum preventable fractions, on how frequently each model is able to generate simulated trials compatible with the observed vaccine efficacy”.
- On the other hand, we think it is important to acknowledge that, with the amount of data at hand, our analysis must be taken with caution, since they are hardly conclusive of what are the exact vaccine mechanisms present in a vaccine such as M72/ASO1E, since, beyond model 5, we also see other models featuring high posteriors, and the posteriors of some of the models may be highly sensitive to relatively small changes in either VE_{dis} or their estimated maximum fractions of preventable cases. Because of these reasons, we have decided to drop the discussion about the possible implications of finding different mechanisms present, or absent, from the top-posterior model case, since they were admittedly too speculative.
- Finally, in our original submission, we commented that integrating a larger number of participants in the trials across study sites or age strata would allow stratifying these analyses in participants’ subgroups, increasing the resolutive potential of these analyses. We now include an extended discussion of these questions in lines 527-539 392-422 in the document with highlighted changes.

We believe that, after introducing these corrections in response to the reviewer concerns, we now submit a more pertinent interpretation of the results found, which is at the same time more robust, clearer and less speculative. We would like to thank the reviewer for their extremely valuable feedback here.

15. L215 “These results evidence the importance of removing arbitrariness from modelling choices of vaccine descriptions”. This is undoubtedly true. but until the authors explain mechanistically *why* the model scenarios and the data have led to results we have seen, this paper has yet to provide good evidence that the “arbitrariness has been reduced”. As elsewhere, authors need to explain mechanistically why the confrontation between model and data has led to structures 5 and 6 being preferred over the other structures. If plausible, this may lead reader to believe that plausible evidence has been created (and the results are not just model artefact/quirk) >> Add explanation

We agree. Following reviewer's advice, we have included further explanations of why, and how, different models receive different posterior probability scores (see lines 280-334 and 435-525 in the document with highlighted changes).

Ultimately, the reason why we do stand behind that claim is that we are just comparing two different procedures, the first of which consists of choosing a general vaccination framework (in our case, of all-or-nothing vaccines), and within it, parametrize a vaccine choosing a given combination of protection mechanisms defined a priori, with no concern on its plausibility in the light of trials' data. Instead, what we propose is to acknowledge all possible model descriptions within the broad vaccination framework chosen, and, produce quantitative estimates for the compatibility of each model with trials' results. By combining all the impact forecasts produced from each of these possible models into a unique weighted average that uses these posteriors as weights, we produce a less arbitrary impact forecast, for it does not rely on a priori-made assumptions of the mechanisms at place. Clearly, this contribution only constitutes a relatively small step towards guaranteeing unbiased evaluations of TB vaccines, and more data in bigger trials conducted in different geographical areas with diverse TB demographics are needed to that end, as we know highlight in lines 527-538).

16. L217. Interestingly, I notice that the simpler vaccine protection structure represented in structure '7' (vaccine prevents all routes equally) is the closest of all the 7 structures to the results from this complicated analysis, two-thirds of the time (ie in India and Indonesia it ranks first, and it ranks third in Ethiopia) (Fig 4D). As such the authors should really say, that if a simpler representation of vaccine effect was needed (because, eg, of computational resources or time limitations or unavailability of code for this new method), using structure 7 may be a good choice. Further, I notice that the authors say that structure 7 is the structure used in the modelling papers highlighted by the authors (refs 25 and 26). As such, this very elegant new work seems to suggest that the results from the simpler work (25 & 26) may be more robust than other papers to this potential source of bias. >> It would be good to make these points in the manuscript.

This is correct, and it implies, as suggested by the reviewer, that the forecasts made in the mentioned previous works would not be incurring into relevant bias in this particular case. However, it is equally important to highlight that this does not guarantee that model 7 could be generally considered less prone to bias than other models, for the situation could perfectly be different for other vaccines, or even for this same vaccine after more efficacy data becomes available. This discussion is now included in the manuscript, as suggested by the reviewer (lines 541-551).

17. L 281 "the potential of our approach to disentangle specific vaccine mechanisms with better specificity than what is presented here could be further exploited if applied to the analysis of multi-centric RCT with divergent TB demographics and, unlike the example analyzed here, homogeneous representation across sites" >> This is really important but not clear enough yet to be useful. The authors need to say what they mean by "divergent TB demographics" and "homogeneous representation across sites". What exactly should a trialist do to provide the information needed?

We thank the reviewer for pointing this out. This discussion has now been added to the manuscript in lines 534-538: "Should further efficacy data for this or other vaccines be collected, based on trials where a minimum number of participants per site and age strata is prioritized (additionally to prioritizing a minimum global number of participants across sites and age groups) this type of Bayesian approach could be extended to integrate more than just one

efficacy observation. This, in turn, would unlock the estimation of more decisive Bayesian posteriors for the different vaccine models proposed.”

17. L 293 “This was done in this study by assuming unbiased representation of the age groups in each country in the study cohorts, and assuming that the overall epidemic risk in the countries of the trial were representative of the overall situation in each specific setting in the year of the study”. >> This assumption is plainly not true as the trial populations were selected to be higher risk than general populations, to decrease sample size. Can the authors speculate here on the likely bias this might introduce to their findings and conclusions?

Actually, we thank the reviewer for noticing this, for the description in the main text of what we actually do was misleading. What we did, and we now describe more precisely in lines 557-561 was “adapting the coarse granularity of participants’ age groups reported in the trial [9,10] (participant numbers were reported for three broad groups of 18-25, 25-30 and 30-50 years old) to the smaller age groups used in the model (seven 5 years-wide age strata: from 15-20 to 45-50) after assuming unbiased representation of the smaller age groups in each country in the wider cohorts reported in the trial.

Beyond these details, we also discuss in lines 565-614 in the resubmitted text why we do believe that our model-based estimates of epidemiological risks are unlikely to introduce significant bias towards an underestimation of epidemiological risks in the settings under analysis.

Minor

18. L186 “These expected efficacy estimates illustrate a sensible feature of our model approach, namely, that a given PoD readout $[\text{VE}]_{\text{dis}}$ must be mapped to lower intrinsic efficacy ϵ values when the vaccine is able to halt progression to disease through all possible routes than when it acts on a subset of them” >> This could have been said before any modelling is done. I suggest it is phrased as a “As expected, ...”

We agree, and we rephrased the sentence added “As expected”, as suggested.

19. L 310 “especially for the analysis of balanced multi-centric trials ” >> What does ‘balanced’ mean here?

Here, we meant that participants are more homogeneously distributed across sites, enabling an analysis stratified per-site. This has been clarified in the text (line 629-637).

20. L241 Discussion. This para is really interesting, but not essential to the value of the work, so could be cut/reduced to make room for the additional material I have suggested (if there is a word count issue)

We have followed the suggestion and dropped most of this discussion, which is now more pertinent given the increased size of the revised manuscript, and cut this part from the discussion.

Clarity

Major

21. Line 23 “unbiased framework ”. This is a big claim, and not at all proven >> Suggest change to “potentially less biased”

We agree with the reviewer. We have completely rewritten the abstract in response to question #11, also considering the abstract size limit in the journal; the “unbiased framework” claim has now been removed.

Minor

22. Line 53 - BGC revax not included in list of vaccines here, but is in the able >> Add bcg revax to text

We agree with the reviewer, and accepted their suggestion (line 51)

23. Line 53 “Remarkable” odd word choice as some of these design choices were very sensible eg testing bcg revax in IGRA-s (as thought only to work in IGRA-s) >> Drop word?

We have accepted the reviewer’s suggestion.

24. Line 89 “Applying our Bayesian formalism, we show that not all possible vaccine mechanisms that a priori could be included in All-or-Nothing compartmental models -arguably the most widely used type of vaccine models used in the modeling literature [17-19]- are equally backed-up by this specific trial readout” >> This is a very complicated sentence. Most of the middle can be cut

We agree with the reviewer, and rephrased that sentence: see lines 115-120.

25. Line 91 “... -arguably the most widely used type of vaccine models used in the modeling literature [17-19]...” >> If this is not dropped, it needs to be justified - state how many of the TB vx papers make this assumption, and how many do not. (the review says there are about 30, so it is not a big task to find out)

We agree with the reviewer, we simplified this sentence and dropped that claim, see lines 115-120.

26. Line 106 “Vinnickys ” Incorrectly spelt

This is true, and it is now corrected. We thank the reviewer for pointing this out.

27. Line 176 “in spite of this model architecture being the one used in most recent modeling works aimed at producing impact estimates for TB vaccines similar to M72/AS01E [25,26]. >> Authors need to justify this. How many of the models in the lit use the various mechanisms the authors describe? Again the lit is small and has been reviewed, so easy to come up with this data to support the statement.

Although we agree with the reviewer in that, in order to backup this claim, we should complete a bibliographic revision of the available literature, it is not our goal here to provide with such review. We have therefore opted for removing this sentence.

Motivation/research question

Major

28. Given the work is presenting a framework that could be used by others, I would expect the model code with documentation to be available. >> Share and document on (eg github)

We definitely agree, and we have now uploaded the whole set of relevant codes, including the full spreading model to Github, which expanded the provided codes in the first version of the manuscript that only covered the scripts necessary to simulate the clinical trials (figure 3A). All relevant codes to the study are openly available at:

https://github.com/MarioTovarCalonge/Bayesian_Framework_TB_Vaccines

where also master scripts and software documentation is provided.

Lit review/framing

Major

29. Line 73-6. "Yet, a majority of the models in the current literature on TB vaccine modeling base their forecasts on assuming that efficacy readouts unequivocally map onto specific combinations of action mechanisms without providing a plausible justification of this important modeling choices [17]." This is true, but this review paper cited does not make this statement. >> Drop sentence, or if the authors want to refrain it, this statement needs to be justified. Eg a summary of which papers were in the 'majority' and which were not.

Again (see point 27 above), we agree with the reviewer on this matter, but it is not our intent to provide with a rigorous bibliographic review of the modeling literature on these particulars. We have therefore dropped this sentence.

Reviewer #2 (Remarks to the Author):

1. This interesting manuscript takes a Bayesian modelling approach to the evaluation of TB vaccine efficacy, using data generated from the M72 vaccine efficacy trial. The authors are correct that although we can determine if a vaccine is effective in an efficacy trial, we cannot determine easily by which mechanism such efficacy is achieved. The authors suggest reasons why such an understanding might be important, which relate to real world efficacy in preventing transmission and real world effectiveness.

The paper is generally well written and explains the concepts clearly. The assumptions – ie that the efficacy observed with M72 / AS01e, may be because of 3 possible scenarios in each latently infected subject enrolled – are correct.

We thank the reviewer for their general positive opinion about our work, and appreciate the fact they agree with our initial hypotheses.

2. This modelling leads to some very interesting conclusions. E.g. the vaccine model offering the highest posterior probabilities is model 6, where vaccine PoD leans on protection against endogenous reactivation of LTBI and TB upon reinfection – I'm not quite sure how the authors extrapolate from this to conclude that the trial result was unlikely to be do to an effect of the adjuvant alone.

We thank the reviewer for pointing out that the way we introduced and justified this conclusion in the light of our analysis was not clear enough; a criticism that we agree with. Being critical about this question, we have to admit that this discussion was probably too speculative, and have, consequently, dropped the discussion from the resubmitted version of the article, also considering that, after correcting some issues in response to reviewer 1 comments, the two first models in the rank of highest posteriors have shifted.

In the resubmitted version of the manuscript, the interpretation of the observed ranking for model posteriors is included in lines 280-334. Essentially, we see that models protecting against only one route to TB disease are less likely to be able to explain, by themselves, the level of protection observed in the M72/ASOE1A. Instead, models including either two (models 4, 5 and 6), or all three mechanisms, present larger posteriors when these are evaluated at the observed estimate of $VE_{dis}=49.7\%$. In the new text, we now emphasize that these posteriors can be used as quantitative estimates of the relative reliability of each model's performance, (allowing us to build a Bayesian, mechanism-agnostic forecast by averaging the forecasts emanating from each model using the Bayesian posteriors as weights), but that, if we want to infer the precise mechanisms at place in a vaccine, more data is needed (see lines 527-539).

3. The 3 countries chosen to illustrate the point have different rates of LTBI in their populations – was this factored into the model in any way?

Yes, the reviewer is correct, the countries here studied have different rates of LTBI in the population, which in turn depends on the different age strata considered, and evolves with time, something that is captured in our model.

Specifically, our model calibrates overall infectiousness and diagnosis rates to sigmoid functions in order to fit overall incidence and mortality trends in each country independently. As a consequence, the size of the LTBI reservoir produced by the model is different across countries and age groups, as it is the relative fraction of infected individuals who are fast progressors (F)

with respect to the total amount of fast progressors plus LTBI (F+L), as it is now included for the three countries of the study in figure 1D.

The procedure consists of the following steps. First, we calibrate the transmission model to fit the incidence and mortality WHO data of the 3 countries involved in the multi-centric clinical trial under analysis, and then we use these model-based fits (which reproduce differences in the size of LTBI (L) and F reservoirs) to obtain estimates of forces of infection, and F vs L fractions in the entire population of each country (figures 1D and 1E in the new submission). Then, these F vs L inferred fractions are used to sample participants iteratively, and stochastically, in each of the instances of the simulated clinical trials. On these sampled sets of trial participants, we simulate trials in a way that captures the expected distribution of Fast vs. slow progressors that are observed in the general population. From these simulated trials, we infer model posteriors and expected efficacy levels under each vaccine description, and generate vaccine impact forecasts.

The entire procedure has now been described in detail in the SI appendix. Code to enable reproduction of all results presented (and not just the trial simulations) has also been added to the Github repository, that now includes all the relevant code in response to this, and other comments by reviewer #1, ensuring transparency and reproducibility.

4. The limitations section contains some logistically implausible suggestions – e.g. that a year prior to enrolment subjects are recruited so the stability of their latent Mtb infection is determined. Whilst scientifically I can see this would be interesting, the logistics and cost would preclude this happening. Unfortunately our ability to accurately distinguish between long term and recent infection is far from established at the moment.

We in general agree with the reviewer's observation, and, as such, we have modified the original statements on that matter in the discussion sections to acknowledge this explicitly (see lines 621-624). Interestingly enough, Tom Scriba, Elisa Nemes and collaborators have recently proposed a new biomarker based on comparing HLA-DR expression levels between $IFN\gamma^+$ and TNF^+ M. tuberculosis-specific T cells and total CD3 T cells (see Ref 27 in the resubmitted manuscript). We reckon that this kind of approaches, paired with enhanced clinical trial design principle and analysis strategies may translate into much more reliable and precise vaccine characterizations.

REVIEWERS' COMMENTS

Reviewer #1 (Remarks to the Author):

This is my re-review of this paper.

The authors have done a fantastic job of responding constructively to the comments and really should be commended on the thoroughness of their response. They have even gone further than suggested to improve the clarity and utility of the work, for example, by toning down the distracting speculation in the discussion.

I have a few small comments below, but nothing major. (page numbers refer to the tracked-changes-showing version)

* Abstract. The abstract is very much improved. Up to the authors of course, but I would suggest changing:

a) "... found that most plausible models for this vaccine include protection against ..." to "... found that most plausible models for this vaccine needed to include protection against ..."

b) Replacing the last sentence with something like "Collecting new data on the impact of TB vaccines in different epidemiological settings would be very useful to improve our model estimates of mechanism." This is a more important take away I think.

* line 73 - suggest dropping 'exhaustive'. Not a particularly useful

* Fig1.

In figure heading

- for panel [B] Say what $p+$ $p-$ np are.

- For Panels CDE: Say what error bars represent

Panel D x-axis title: % of what. What is the denominator here? Add to the x-axis title

Panel E x-axis title: add the force of infection units. Add to the x-axis title

Panel F x-axis title: Fraction of what? Presumably Fraction of all TB cases per year? Add to the x-axis title

Richard White

Reviewer #2 (Remarks to the Author):

The authors have addressed the reviewers comments

REVIEWER COMMENTS

Reviewer #1 (Remarks to the Author):

This is my re-review of this paper.

The authors have done a fantastic job of responding constructively to the comments and really should be commended on the thoroughness of their response. They have even gone further than suggested to improve the clarity and utility of the work, for example, by toning down the distracting speculation in the discussion.

We would like to thank sincerely the reviewer for his comments through the review of the manuscript, for his feedback has helped us greatly to improve the overall quality of the work.

I have a few small comments below, but nothing major. (page numbers refer to the tracked-changes-showing version)

*Abstract. The abstract is very much improved. Up to the authors of course, but I would suggest changing:

a) "... found that most plausible models for this vaccine include protection against ..." to "... found that most plausible models for this vaccine needed to include protection against ..."

We have updated the text as suggested.

b) Replacing the last sentence with something like "Collecting new data on the impact of TB vaccines in different epidemiological settings would be very useful to improve our model estimates of mechanism." This is a more important take away I think.

We have updated the text as suggested as we agree with the reviewer.

* line 73 - suggest dropping 'exhaustive'. Not a particularly useful

This is now updated.

* Fig1.

In figure heading

- for panel [B] Say what $p+$ $p-$ np are.

- For Panels CDE: Say what error bars represent

Panel D x-axis title: % of what. What is the denominator here? Add to the x-axis title

Panel E x-axis title: add the force of infection units. Add to the x-axis title $t(-1)$

Panel F x-axis title: Fraction of what? Presumably Fraction of all TB cases per year?

Add to the x-axis title

We have updated the Figure 1 caption and panels d,e and f axis to address reviewer's comments.

Richard White

Reviewer #2 (Remarks to the Author):

The authors have addressed the reviewers comments

We thank the reviewer for their general positive opinion about our work, as well as for their suggestions and comments that were very helpful to improve the quality of our manuscript.